# Diamond preservation in the lithospheric mantle recorded by olivine in kimberlites

Andrea Giuliani [1], David Phillips [2], D. Graham Pearson [3], Soumendu Sarkar[2], Alex A. Müller [3], Yaakov Weiss [4], Robin Preston[5], Michael Seller[6] & Zdislav Spetsius[7]

The diamond potential of kimberlites is difficult to assess due to several mantle and magmatic processes affecting diamond content. Traditionally, initial evaluations are based on the compositions of mantle-derived minerals (garnet, chromite, clinopyroxene), which allow an assessment of pressure-temperature conditions and lithologies suitable for diamond formation. Here we explore a complementary approach that considers the conditions of diamonds destruction by interaction with melts/fluids (metasomatism). We test the hypothesis that carbonate-rich metasomatism related to kimberlite melt infiltration into the deep lithosphere is detrimental to diamond preservation. Our results show that high diamond grades in kimberlites worldwide are exclusively associated with high-Mg/Fe olivine, which corresponds to mantle lithosphere minimally affected by kimberlite-related metasomatism. Diamond dissolution in strongly metasomatised lithosphere containing low-Mg/Fe olivine provides a causal link to the empirical associations between low diamond grades, abundant Ti-Zr-rich garnets and kimberlites with high Ti and low Mg contents. This finding show-cases olivine geochemistry as a viable tool in diamond exploration.

Diamond deposits are predominantly hosted by complex volcanic rocks of mantle origin, namely kimberlites (-70% of world production by value) and, to a lesser extent, olivine lamproites (5%), with the remaining diamond production coming from alluvial deposits[1]. Kimberlites (and lamproites) are considered to be the transporting agents for diamonds, as diamonds are only stable at high pressures (>4 GPa or >130 km for continental geotherms; Fig. 1) and represent xenocrysts sourced mainly from the lithospheric mantle. This notion has led to the widely held view that the composition of kimberlites has no relationship to their diamond cargo[2]. However, a few studies[3–5] have suggested a potential link between kimberlite chemistry and diamond grade (i.e., concentration).

Very few kimberlites (-1% of those discovered to date) contain diamonds in suitable abundance and quality to be mined profitably.

Evaluating the diamond potential of a kimberlite is challenging and typically requires expensive bulk sampling of kimberlites. This is because diamonds are extremely rare in even the most diamondiferous kimberlites and occur at part per million (ppm) or, more commonly, sub-ppm concentrations[6]. In addition, the concentration (or grade) and quality of diamonds can vary widely within an individual kimberlite locality[7,8]. Consequently, alternative (or complementary) approaches to diamond deposit evaluation are desirable before undertaking bulk diamond sampling.

Based on studies of mineral inclusions in diamonds, it is now well established that the majority of diamonds (>90%) entrained by kimberlites are of lithospheric origin, with the remainder coming from convecting mantle sources[9]. This observation has led to the definition

[1]Institute for Geochemistry and Petrology, Department of Earth Sciences, ETH Zürich, Zürich, Switzerland. [2]School of Geography, Earth and Atmospheric Sciences, University of Melbourne, Melbourne, VIC, Australia. [3]Department of Earth and Atmospheric Sciences, University of Alberta, Edmonton, AL, Canada. [4]Institute of Earth Sciences, The Hebrew University, Jerusalem, Israel. [5]De Beers Group, Johannesburg, South Africa. [6]De Beers Group, Toronto, ON, Canada. [7]Institute of Diamond and Precious Metal Geology, Siberian Branch of the Russian Academy of Science, Yakutsk, Russia. ✉ e-mail: andrea.giuliani@erdw.ethz.ch

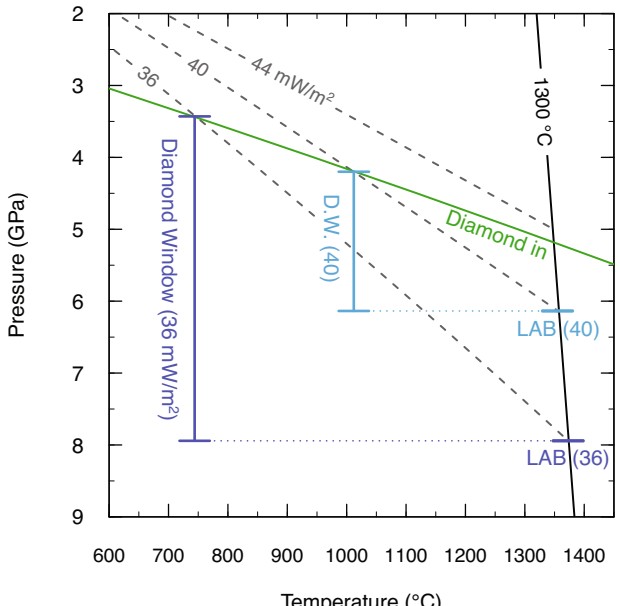

**Fig. 1 | Pressure-temperature covariation diagram showing the relationship between lithospheric-mantle conductive geotherms, lithospheric thickness and diamond window.** The diamond window is here defined as the depth interval bracketed by the intersection of conductive geotherm and graphite-diamond transition ("diamond in"), and the intersection of conductive geotherm and mantle adiabat (here assumed to be 1300 °C). A cold lithospheric-mantle geotherm (e.g., 36 mW/m²) corresponds to a deeper lithosphere-asthenosphere boundary (LAB) and thicker diamond window. The diamond window is absent for much hotter conductive geotherms (e.g., 44 mW/m²), which do not intersect the diamond stability field. Conductive geotherms from Hasterock and Chapman[57].

of the "diamond window" corresponding to the depth interval between the graphite-diamond phase transition and lithosphere-asthenosphere boundary (LAB; Fig. 1). Thicker diamond windows may be associated with higher diamond grades due to more prolonged diamond entrainment by traversing kimberlite magmas[10,11]. In addition, studies of diamondiferous mantle xenoliths and mineral inclusions in diamonds have demonstrated that, with some exceptions (e.g., lherzolites from the Victor mine, Canada), diamonds are associated with particular mantle lithologies, namely highly depleted garnet and/or spinel harzburgites (and dunites) and high-pressure eclogites[9,12–14]. The common association of diamond with highly depleted substrates is confirmed by the high Mg# [molar proportions of Mg/(Mg+Fe)×100] of olivine inclusions in diamonds (predominantly >91; ref. 9). Conversely, strongly metasomatised lithologies enriched in Fe and Ti are rarely associated with diamonds. Therefore, understanding the pressure, temperature and compositional features of lithospheric mantle material entrained by kimberlites is fundamental to diamond exploration[2,6]. These constraints can be determined by studying the composition of kimberlite indicator minerals such as garnet, chromite, clinopyroxene, ilmenite and, as shown below, olivine. Indicator minerals are xenocrysts released, together with diamond, during disaggregation of mantle wall rocks by the transporting kimberlite magmas. The aim of indicator mineral studies is to detect the presence of diamond-bearing environments in the lithospheric mantle traversed by kimberlites, including the relative proportions of lithologies and their depth distribution[2,10,15].

Typical indicator mineral approaches to diamond exploration focus on chemical conditions that favour the presence of diamond. Here we explore a complementary approach that examines the diamond tenor in kimberlites by tracking conditions that result in diamond dissolution in the lithospheric mantle. This strategy builds on empirical observations that certain compositional features, such as

elevated Ti, Zr and Y concentrations in garnet and high $Fe^{3+}$ in ilmenite recovered from kimberlites, which are attributed to intense melt metasomatism and increasing oxygen fugacity in the lithospheric mantle, are commonly associated with poor diamond preservation and low diamond grades[2,3,10,13,15,16]. However, the exact cause of the correlation between low diamond preservation and "Ti-rich" melt metasomatism is unclear, as is the composition and origin of the melt(s) involved, the timing of metasomatism, and their potential relationship to kimberlite magmatism.

In this study, we use the compositional systematics of olivine in kimberlites to infer mantle metasomatic conditions and diamond preservation potential. We specifically test the hypothesis, based on previous observations of diamond resorption features[17,18] and recent experimental work[19,20], that interaction of diamonds with carbonate-rich kimberlitic melts at lithospheric mantle depths can reduce the diamond tenor of the lithospheric mantle, potentially rendering kimberlites uneconomic. Olivine in kimberlites has been shown to record the extent of kimberlite-related metasomatism in the lithospheric mantle[21–23] and provides a practical means to test this premise by comparing olivine compositions in various kimberlites with their published diamond grades.

Below we show that high diamond grades are never associated with kimberlites containing olivine with mean Mg# lower than 89 and 90 for magmatic and xenocrystic olivine, respectively. Mantle-derived olivine with Mg# lower than 89–90 is linked to lithospheric mantle substrates affected by metasomatism due to the infiltration of early pulses of kimberlite (or related) melts, thus correlating diamond destruction with kimberlite-related metasomatism. This finding provides a direct link between the composition of kimberlites and their diamond content, and highlights a viable, cost-effective and practical method for diamond exploration.

## Results and Discussion
### Comparison of olivine geochemistry with diamond grades
Olivine, the main constituent of fresh kimberlite rocks[24], is generally zoned between xenocrystic cores derived from the disaggregation of lithospheric mantle wall rocks, and magmatic rims[25] (Fig. 2). Olivine cores do not simply represent typical cratonic mantle peridotites, but may also be locally sourced from other lithologies (e.g., megacrysts; sheared peridotites) that have experienced metasomatism by precursor kimberlite melts[23,25–27]. It is now well established that the eruption of kimberlite magmas is preceded by 'priming' of lithospheric mantle conduits from earlier failed pulses of kimberlite melt[23,25,28–30]. The average Mg# of xenocrystic olivine cores is directly correlated with the average Mg# of olivine rims in kimberlites on global[21] and regional scales[31,32]. This correlation, combined with extensive petrographic and experimental evidence of assimilation of entrained lithospheric mantle material by kimberlites[29,33–35], suggests that the composition of kimberlite melts at surface is directly related to the composition of lithospheric mantle wall rocks, which interact with kimberlite melts en route to surface[21]. In addition, more assimilation of metasomatised material yields more Fe and Ti-rich (and Mg-poor) magmatic olivine and kimberlite groundmass[22,36]. Therefore, olivine geochemistry provides a direct link between kimberlite melt composition and traversed lithospheric mantle wall rocks, including the extent of kimberlite-related metasomatism.

For this study, we assembled an updated olivine compositional database for kimberlites worldwide, including additional electron microprobe analyses (see Methods) of olivine from 13 kimberlites in Russia, Canada, Brazil and South Africa. This dataset (Supplementary Data 1) confirms that the average Mg# of olivine cores in kimberlites (and also diamondiferous cratonic lamproites; see Sarkar et al. [37]) is directly correlated with the average Mg# of olivine rims (Fig. 3). Of the 100 localities for which olivine data are now available, 79 have available diamond grades (e.g., run of mine; exploration data) or are barren

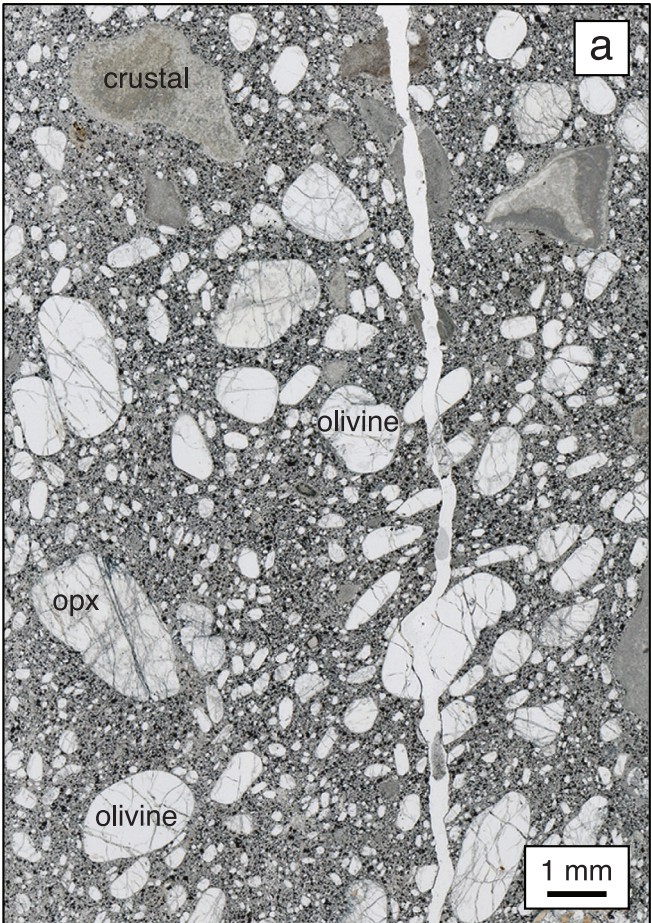

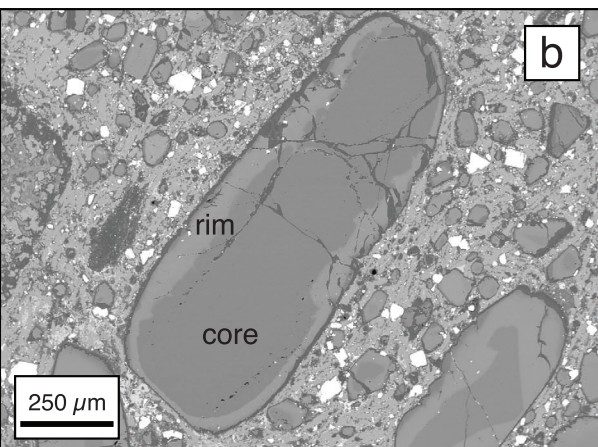

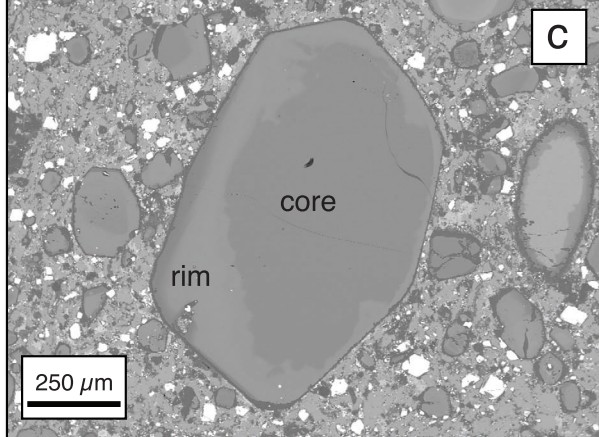

Fig. 2 | **Abundance and zoning of olivine in kimberlites. a** Thin section photomicrograph from the Internationalnaya kimberlite (Siberia) and **b**, **c** back-scattered electron (BSE) images of representative zoned olivine grains. Note the abundance and size-shape variability of olivine. The sample also includes orthopyroxene ('opx') and crustal xenocrysts.

(Supplementary Data 1). Comparison of the average Mg# of either olivine cores or rims (which are linearly correlated) with diamond grades indicates that high diamond grades (≥50 cpht or carats per hundred tonnes; $n = 28$) are exclusively associated with Mg# ≥90.3 (and CaO ≤0.05 wt.%; with one exception at 0.06 wt%) for olivine cores and ≥89.0 for olivine rims (Fig. 4; Supplementary Figure 1). Conversely, low diamond grades (≤22 cpht) are characteristic of kimberlites containing lower-Mg# olivine (cores ≤89.5, $n = 18$; or rims ≤88.3; $n = 23$). However, low diamond grades can also occur in kimberlites with high-Mg# olivine (i.e., cores >90.3, $n = 15$; or rims >89.0, $n = 12$). In addition, kimberlites containing abundant (relative to their production) sub-lithospheric diamonds (e.g., Karowe in Botswana; Letseng in Lesotho; Monastery and Jagersfontein in South Africa) show the same correlation between olivine Mg# and diamond grade.

### Diamond resorption by kimberlite melts in the lithospheric mantle

The observed correlation between olivine core compositions and diamond grades in kimberlites (Fig. 4a,c) is consistent with our hypothesis that increasing kimberlite-related metasomatism of the deep lithosphere reduces diamond abundance in entrained mantle wall-rocks, while decreasing the Mg# and increasing Ca contents of entrained olivine xenocrysts. These combined effects translate to lower Mg# in the rims of magmatic olivine (Fig. 3) and lower diamond grades in kimberlites at surface (Fig. 4). The detrimental effect of kimberlite-related metasomatism on diamond preservation in the deep lithosphere provides a causal mechanism to explain the empirical

inverse correlation between high Zr and Ti in garnet (sourced within the diamond window) and diamond grades reported in previous studies[10,15]. Garnets with high Zr and Ti are typical of sheared peridotites[10], considered to have formed by interaction of cratonic peridotites with kimberlite melts[30]. Similarly, lower diamonds grades in younger kimberlites in regions of long-lived kimberlite (and lamproite) magmatism (e.g., Barkly West, South Africa; Rankin Inlet, Canada) have been attributed to diamond destruction by the protracted infiltration of earlier kimberlite or related melts[38]. Studies of diamondiferous and barren eclogites from the same localities have also shown that interaction with high-Ti carbonate-rich melts, potentially related to the kimberlite host, eliminates diamonds[39]. This metasomatic style differs from the low-Ti carbonatitic metasomatism which, based on studies of fluid inclusions in diamonds, is commonly invoked to form diamonds in the lithospheric mantle[40]. Our findings suggest that the interaction of diamonds with kimberlite (or related) melts in the deep lithosphere can result in partial to almost complete dissolution of diamonds depending on the intensity of metasomatism. This process is expected to impact diamondiferous peridotite and eclogite substrates equally because localities with dominant peridotitic or eclogitic diamonds are present in the high-, moderate- and low-grade groups (see Supplementary Data 1).

Diamonds recovered from kimberlites often exhibit resorption features associated with progressive rounding of original octahedral shapes and/or surface dissolution features[17,18]. The bulk of this resorption is attributed to resorption during transport in the kimberlite medium[17,18,38,41]. However, the above lines of evidence indicate that

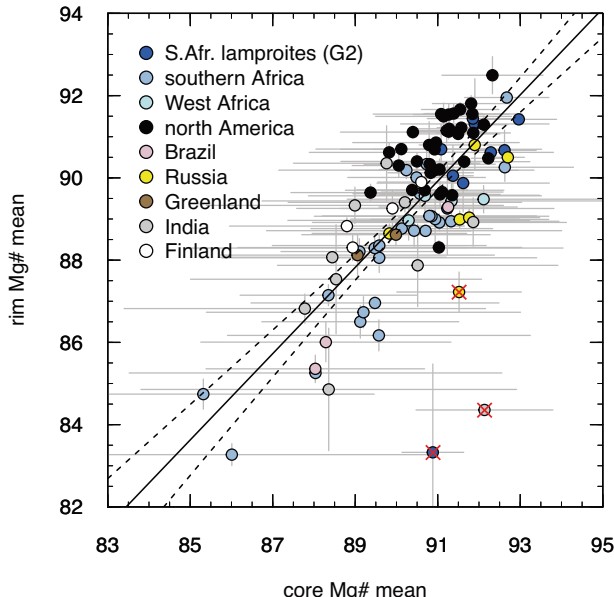

**Fig. 3 | Correlation between mean olivine core and rim compositions in kimberlites worldwide.** Olivine lamproites from South Africa and India are also included. Mg# = Mg/(Mg/Fe) as molar proportions. The black line shows a regression through the data and dotted lines bracket the related 1σ confidence interval. Red crosses indicate outliers representing magmas that may have lost part of their cargo of metasomatised (i.e., Fe-rich) xenocrystic cores[23]. The error bars represent 1 standard deviation of the mean.

diamond dissolution can also occur in the mantle source, potentially related to 'priming' of the lithosphere by early carbonate-rich kimberlite melts. This argument is supported by rounding of diamonds inside mantle xenoliths, where the diamonds were protected from interaction with the transporting kimberlite[42,43], and surface features consistent with kimberlite transport overprinting pre-existing rounding inherited from lithospheric processes[18]. Some dissolution features observed on the surface of diamonds might also indicate interaction with carbonate-rich melts in the deep lithosphere[19,20].

Our results are consistent with locally intensive dissolution of diamonds in the lithospheric mantle before kimberlite entrainment, based on the scarcity of diamonds in lithospheric mantle columns which have been extensively modified by infiltrating kimberlite melts (i.e., mean Mg# and CaO in olivine cores <90 and >0.06 wt.%, respectively; Fig. 4c). In this regard, it is noteworthy that kimberlites with very high-Mg# olivine from Lac de Gras, Canada (core mean Mg# = 89.4−92.1; n = 26) and Siberia (89.8−91.8; n = 6) plus lamproites from the West Kaapvaal craton, South Africa, generally show substantially higher proportions of (unresorbed) octahedral diamonds compared to Cretaceous southern African kimberlites with lower Mg# (85.3-91.3; n = 12) in olivine[18,41]. However, in kimberlites from Wafangdian and Mengyn (China), lower diamond grades correlate with higher relative abundances of octahedral diamonds[44]. A more detailed assessment of the potential relationship between diamond rounding, diamond resorption features, and olivine composition would be required to better understand the full effect of kimberlite-related metasomatism on the quality and therefore value of diamonds, which is another fundamental factor controlling the economics of diamond-bearing kimberlite. Yet, published data on diamond resorption are scarce and only available for a few localities.

### The influence of lithospheric thickness and other processes
While there are currently no kimberlites containing olivine cores with mean Mg# <90.3 and olivine rims with Mg# <89.0, that exhibit high

diamond grades (>50 cpht), high Mg# in olivine does not always correspond to grades >50 cpht (Fig. 4a,b). We have assessed this further by 'scoring' kimberlites based on the combined mean Mg# values of cores and rims (see Methods) and arranging kimberlites in order of decreasing diamond grade (Fig. 4d). This figure shows that the Tonguma kimberlite is the only high-grade locality with a score of 2 [i.e., moderately high Mg# in olivine cores (90.3) and rims (89.0)], which might be attributed to preferential analysis of Fe-rich cores (see Methods). It also shows that 7 kimberlites with high to very high Mg# in olivine (scores of 3 and 4) feature low diamond grades, clearly indicating that lithospheric mantle metasomatism may not be the only factor influencing diamond grades. As noted above, the diamond content of a kimberlite is a function of multiple factors including thickness of the diamond window[10,11] (Fig. 1), occurrence of diamondiferous lithologies and their diamond content[9,12–14], diamond resorption during transport in kimberlite magmas[17,18,45], volcanic sorting, and dilution with crustal country rocks[13,46–48]. Diamond resorption in the lithospheric mantle appears to be an additional key factor that might be locally dominant (kimberlites with core Mg# <90.3 and rim Mg# <89.0), but is not an absolute predictor of diamond grade. In these kimberlites, other factors appear to play an important role.

To gain additional insights into diamond grade variability in kimberlites, we compared diamond grade, lithospheric thickness (see Methods) and olivine rim Mg# for the 55 localities for which these data are available (Fig. 5). Lithospheric thickness can be considered a proxy of the vertical extent of the diamond window (Fig. 1). In this more restricted dataset diamond grades >50 cpht are exclusively associated with kimberlites containing high-Mg# olivine (score 3 and 4) and underlain by lithosphere equal to, or thicker than, ~200 km. Only three kimberlites with less than 50 cpht feature high-Mg# olivine and thick lithosphere (≥200 km): Lace (~34.5 cpht) and Jagersfontein (~9 cpht) in South Africa, and Leslie (~30 cpht) in the Lac de Gras region (Canada). A possible explanation for these exceptions is the originally lower concentrations of diamonds in the lithospheric mantle underlying these kimberlites or, in the Leslie case, substantial dissolution of diamonds in the kimberlite melt during transport[49].

The combination of diamond grades and lithospheric thickness estimates also shows that kimberlites hosting high-Mg# olivine (score 3 and 4) exhibit lower diamond grades where the lithosphere is thinner than 200 km (e.g., Tres Ranchos-04 in Brazil; Peuyuk in Canada; Melton Wold in South Africa) (Fig. 5). Finally, the low grades of some high-Mg# olivine bodies (e.g., Newlands in South Africa; olivine score = 4; grade ~19.4 cpht) (Fig. 4) for which robust constraints on the lithospheric thickness are not available, could be also explained by other processes. In small dykes such as at Newlands and Roberts Victor (~40 cpht; South Africa), only a fraction of the lithospheric cargo might have been transported to surface due to density separation in the ascending melt[47] and/or ore dilution by country-rock material. These results underscore the need to integrate complementary information from different methods to predict diamond grades in kimberlites (Fig. 6).

### An additional diamond evaluation tool
Previous studies have utilised the abundance and size distribution of olivine as a proxy for diamond sorting during volcanic processes, as these minerals have similar hydrodynamic properties[46,48]. Our work reveals that olivine has another major role in diamond evaluation, namely predicting diamond grades. Importantly, this is a rapid, inexpensive tool that only requires major element analyses of several, preferably zoned, olivine grains in a few kimberlite thin sections. The method is considered complementary to other indicator mineral (e.g., garnet, clinopyroxene) geochemical approaches. An important caveat is that olivine compositions are usually relatively invariant in

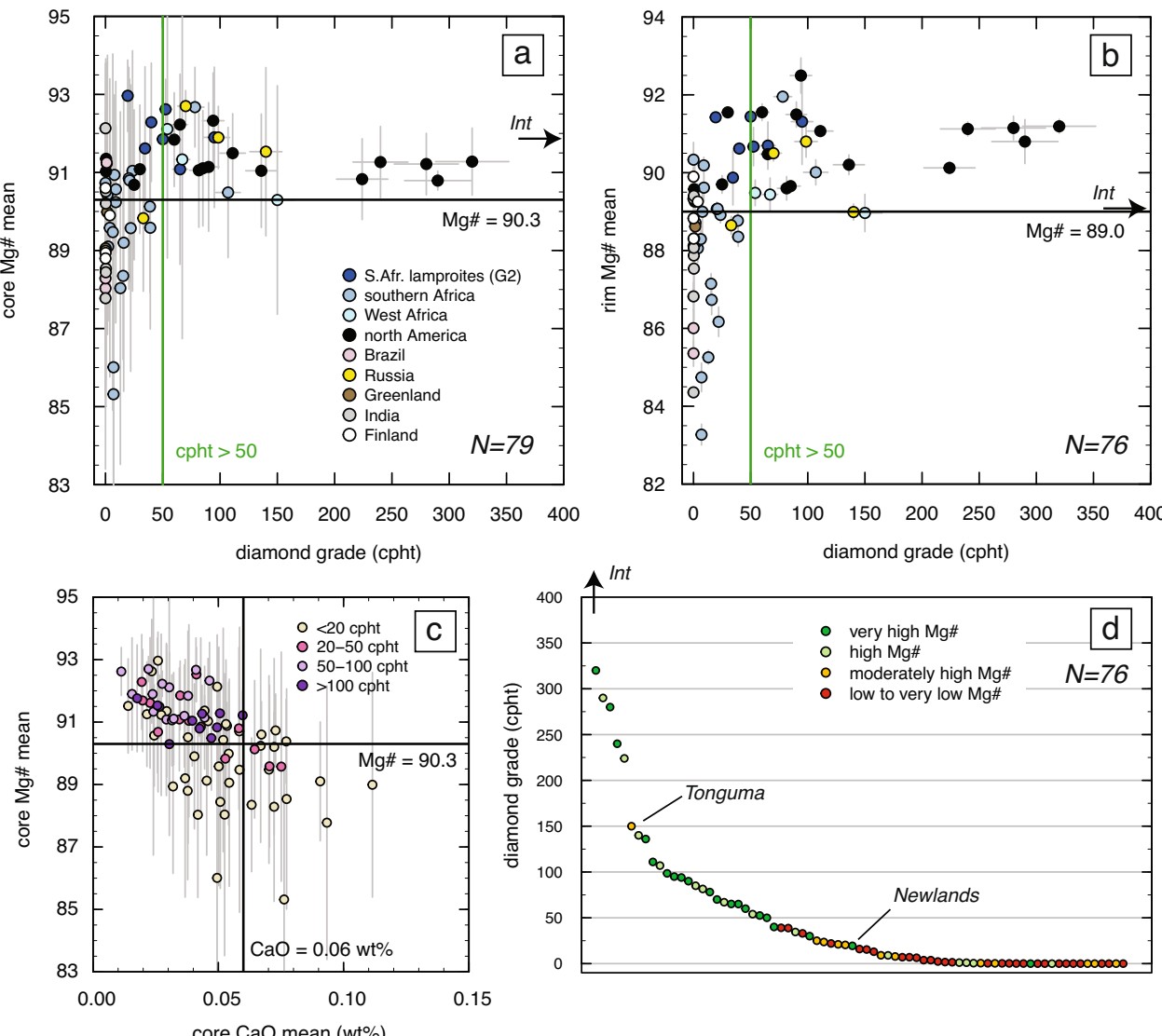

**Fig. 4 | Relationship between olivine composition and diamond grade in kimberlites worldwide.** Olivine lamproites from South Africa and India are again included. **a** Diamond grade (carats per hundred tonnes or cpht) vs mean Mg# (= Mg/(Mg/Fe) as molar proportions) of olivine cores, and **b** olivine rims. The vertical light green line indicates the boundary between high and moderate to low-grade kimberlites, which is set at 50 cpht. The horizontal dark green lines, which corresponds to Mg# of 90.3 for the cores and 89.0 for the rims, indicate the lower limit of kimberlites with high diamond grades (≥50 cpht). The arrow labelled 'Int'

shows olivine compositions from the Internationalnaya kimberlite, which has an estimated grade of 937 cpht (Supplementary Data 1). The error bars represent 1 standard deviation of the mean for olivine Mg#, and 10% of the reported value for diamond grades. (**c**) Mean Mg# vs CaO content for olivine cores colour-coded relative to diamond grade, showing that high diamond grades are limited to kimberlites in the upper left quadrant (Mg# ≥90.3 and CaO ≤0.06 wt.%). (**d**) Plot of kimberlites sorted by diamond grade and colour-coded relative to olivine Mg# score (see Methods). Tonguma and Newlands are exceptions discussed in the text.

kimberlite pipes and clusters of pipes[23,31], whereas diamond grades may vary within volcanic units of the same pipe[7,8]. Hence, olivine compositions provide constraints on the likelihood of diamond preservation in the lithospheric mantle, rather than a direct predicator of actual diamond grade. This is because other local-scale processes may also impact diamond grades, including the original diamond contents in the lithospheric wall rocks, sampling efficiency by kimberlite melts, diamond resorption in kimberlite magmas en route to surface, sorting of entrained mantle material during kimberlite ascent and emplacement, and dilution by country rocks.

A further implication of this study is that the composition of kimberlite rocks and hence their parental magmas have direct bearing on the abundance of entrained diamonds. Olivine is the most abundant constituent in fresh kimberlite rocks[24] and essentially controls the Mg and Fe concentrations in these rocks[50]. Previous

work has also shown that the composition of olivine is directly correlated with the mineralogy of kimberlites, with lower-Mg# olivine occurring in kimberlites that are enriched in Fe-Ti oxide minerals[36]. These combined observations explain why high diamond grades are associated with kimberlite rocks containing high concentrations of Mg and low contents of Ti ([3–5]). This suggests that bulk kimberlite compositions could also be used to evaluate diamond potential, provided that the effects of crustal contamination and hydrothermal alteration are taken into account[51,52]. In summary, kimberlites are not simply vehicles for transporting diamonds from the lithospheric (or deeper) mantle, but provide a wealth of predictive information on the structure and composition of the lithospheric mantle (including diamonds), which can be interrogated to support diamond exploration/evaluation programs using practical, cost-effective geochemical methods.

## Methods

The present compilation of olivine compositions includes datasets previously assembled by Giuliani et al. [21] for kimberlites and Sarkar et al. [37] for olivine lamproites, except for the West Kimberley lamproites which have different compositional zoning compared to the samples included herein. These datasets are augmented with the results from Dalton et al. [32], Tovey et al. [31], Viljoen et al. [53] and new data for samples from 13 additional localities. The latter include Inter-

nationalnaya ($n = 1$), Poiskovaya ($n = 2$), Zapolyarnaya ($n = 3$), Deymos ($n = 2$) and Obnazhennaya ($n = 2$) in Russia; Ghacho Kue ($n = 3$), Victor ($n = 1$), Dharma ($n = 4$) and Chidliak-07 ($n = 3$) in Canada; Leicester ($n = 2$) and Premier ($n = 1$) in South Africa; Karowe ($n = 1$) in Botswana; and Perdizes-04 ($n = 1$) in Brazil. All the samples represent coherent kimberlites (Fig. 2), where olivine was partly to fully preserved allowing measurement of cores and rims in most samples.

Olivine grains for electron microprobe analysis (EPMA) were selected using an optical microscope to target different grain sizes and shapes, while avoiding any bias related to preferential selection of strongly zoned grains[25]. Core and rim selection using the electron microprobe was based on contrasting BSE properties (Fig. 2). Where no contrast was observed, the rim analysis was placed close to the grain edge. Electron microprobe analyses of all the samples, except those from Canada, were undertaken using a JEOL-JXA8530F electron microprobe with five wavelength dispersive spectrometers (WDS) at the University of Melbourne. The analytical conditions were as follows: beam acceleration voltage 15 kV, beam current 20 nA, beam diameter of 2 μm, and counting times per analysis of 20 s on peak positions and 10 s on two background positions located on either side of the peak position. Natural and synthetic materials used for calibrations include wollastonite (Si and Ca), Ti oxide (Ti), Al oxide (Al), chromite (Cr), hematite (Fe), Mn metal (Mn), Mg oxide (Mg), and Fe-Ni alloy (Ni). San Carlos olivine was measured together with the unknowns to assess data quality and reproducibility (Supplementary Data 2). Data reduction included the ZAF matrix correction. Major and minor element compositions of olivine in the Canadian kimberlites were measured at the University of Alberta using a Cameca SX-100. Operating conditions included an accelerating voltage 15 kV, beam current 20 nA, beam diameter 2 μm, and count times on both peaks and backgrounds between 20 and 60 s. The calibrant materials used were a mixture of natural and synthetic minerals, including pure metals and oxides. Further details of these analyses can be found in Sarkar et al. [54].

After data acquisition the grains showing no clear BSE zoning were screened in the following way. If core and rim showed different compositions, they were both retained and classified as mantle-derived 'xenocrystic core' and 'magmatic rim', respectively. If the core and rim analyses were indistinguishable, only the core analysis was retained and classified as either 'xenocrystic core' or 'magmatic rim' after comparison with the analyses of zoned grains. This approach is facilitated by the restricted Mg# range of magmatic rims of olivine in

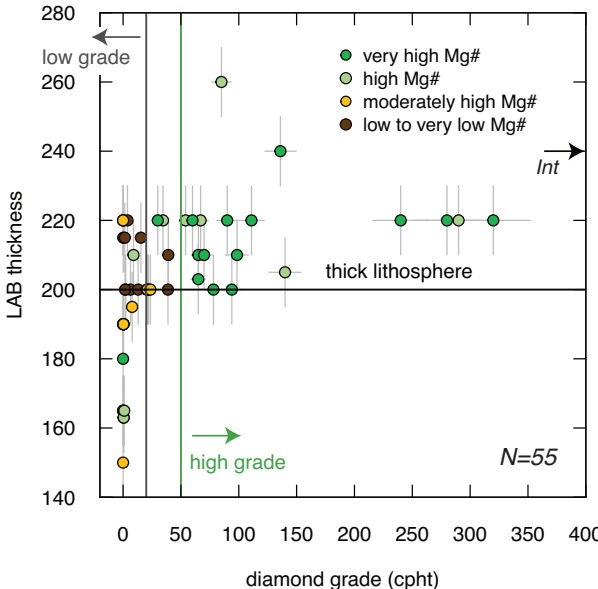

**Fig. 5 | Relationship between lithospheric thickness and diamond grades in kimberlites worldwide relative to olivine composition.** Olivine lamproites from South Africa and India are again included. The error bars represent 10% of the reported value for diamond grades and 10 km for lithospheric thickness. The kimberlites are colour-coded based on olivine Mg# score (see Methods). Note that low grades (<20 cpht) occur in regions with lithosphere thinner than 200 km, regardless of olivine Mg#, and where thick lithosphere (≥200 km) is associated with a low olivine-Mg# score. Conversely, thick lithosphere and high-Mg# olivine are generally associated with high diamond grades (>50 cpht) with some exceptions.

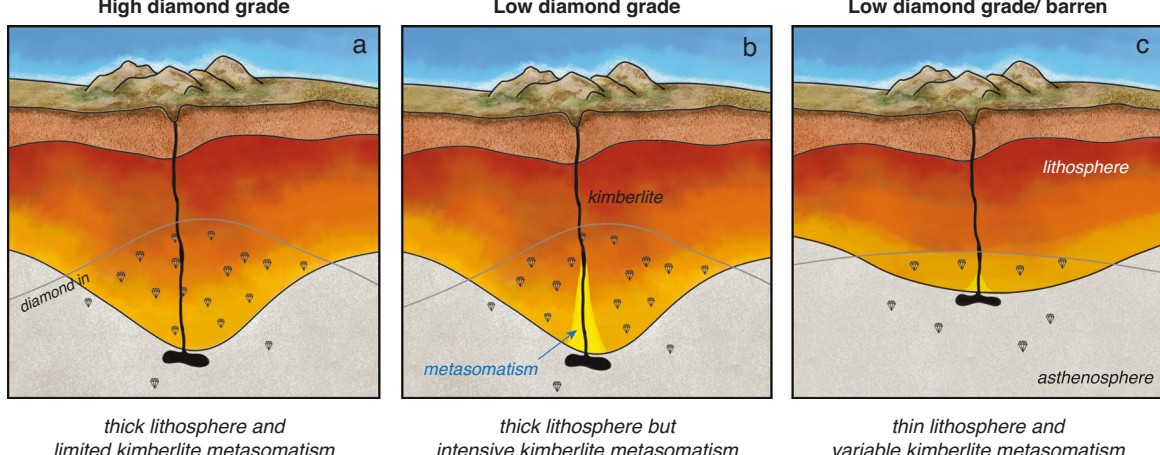

**Fig. 6 | Diamond potential of kimberlites based on lithospheric thickness and extent of kimberlite-related metasomatism. a** High diamond grades are associated with thick lithospheric roots minimally affected by kimberlite metasomatism. **b** Increasing kimberlite metasomatism dissolves diamonds and lowers diamond grades in the ascending kimberlite magmas. **c** Thin lithosphere is associated with low diamond grades regardless of the extent of kimberlite metasomatism.

every kimberlite compared to the large Mg# spread of the cores[25]. The screened data are reported in Supplementary Data 2 and plotted in Supplementary Figure 2.

Average compositions were calculated for cores and rims in each kimberlite and, where available, data from multiple samples were pooled. No substantial difference was observed in olivine compositions from different samples of the same kimberlite, which is consistent with previous studies[23,25,31]. Beyond average core and rim compositions, an 'olivine Mg# score' was calculated for each kimberlite in the following way. For average olivine cores, Mg# values < 90, between 90 and 91, and >91 correspond to scores of 0, 1 and 2, respectively. For olivine rims, Mg# mean values of <89, between 89 and 90, and >90 were assigned scores of 0, 1 and 2, respectively. The olivine Mg# score is the sum of the Mg# scores of core and rim mean compositions (4: very high Mg#; 3: high Mg#; 2: moderately high Mg#; 1: moderately low Mg#; 0: low Mg#). The employment of this olivine Mg# score minimizes potential biases encountered when considering olivine cores and rims separately (e.g., biases due to oversampling of Mg-rich or Fe-rich core compositions such as the Tonguma case) (Fig. 4).

Olivine compositions were then compared to diamond grades obtained from several sources (Supplementary Data 1). These include values reported in previous peer-reviewed publications[1,55] or available from mining company annual reports, and also include new results released by the De Beers Group for the Colossus and Wimbledon kimberlites. The reported diamond grades were determined from a wide range of rock volumes and include run-of-mine values where grades are averaged over one or more years, as well as smaller exploration volumes of kimberlites that have not been mined commercially, to date. Considering the large diamond grade variability observed in every kimberlite[7,8], reported diamond grades embody a large uncertainty that is challenging to quantify. In this study, we have assumed uncertainties of 10% (Fig. 4).

Finally, lithospheric thickness values are based on published estimates (Supplementary Data 1) and correspond to the intersection between the mantle adiabat and xenolith or xenocryst-based geotherms (i.e., curves defining temperature variation with depth). Where possible, lithospheric thicknesses were estimated using the FIPLOT-based approach of Mather et al.[56], which produces best-fit geotherms to mantle-derived pressure-temperature data, using local crustal parameters. We prioritise geotherms determined using well-equilibrated mantle xenolith samples, or databases containing carefully filtered clinopyroxene thermobarometry. Where this approach was not possible, (e.g., most Siberian kimberlites), we used the best approximation of available xenocryst-based pressure-temperature data arrays relative to the geotherms of Hasterock and Chapman[57]; these are broadly equivalent to those calculated by FITPLOT. Because of this varied approach, the uncertainty in calculated mantle temperatures, and the lack of available crustal thickness and heat generation parameters for some locations, the reported lithospheric thickness estimates have an overall uncertainty of at least 10 to 20 km in most cases.

## Data availability
All the data used in this study are provided in the Supplementary Data files.

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

## Acknowledgements

This project was funded in part by De Beers Group, which is also thanked for providing confidential grade estimates for the Colossus and Wimbledon kimberlites. Additional funding was provided by the Swiss National Science Foundation (Ambizione fellowship no. PZ00P2_180126/1 to AG) and NSERC (Discovery grant to DGP). Insightful discussions with Bill Griffin are duly acknowledged. Senan Oesch and Chiara Mattioli are thanked for their help with figure preparation.

## Author contributions

A.G. and D.P. conceived the project and developed the conceptual model. A.G., D.P., D.G.P., Y.W., R.P., M.S. and Z.S. collected the samples for this project. A.G., D.G.P., S.S. and A.A.M. analysed the samples. A.G. developed the model after discussions with D.P., D.G.P., Y.W., R.P. and M.S. A.G. wrote the manuscript with contributions from all the co-authors.

## Competing interests

The authors declare no competing interests.
