## [Peer Review File · Nature Communications]

REVIEWER COMMENTS

Reviewer #1 (Remarks to the Author):

This study reveals an interesting connection between the composition of olivine rims from kimberlites and diamond grade, connecting both to the extent of mantle metasomatism by proto-kimberlitic melts. The authors compiled a very large database to support their proposal and the conclusions are based on examination of this very large set of samples. Rigorous methods are used to eliminate or minimize biases in olivine analyses. The study introduces method of scoring of Mg-number of olivine cores and rims, and then combining them. Fig 4d is a very innovative way of correlating olivine composition with diamond grade. To the best of my knowledge, this is the first study that attempt to quantify the effect of partial dissolution of diamond due to mantle metasomatism on diamond grade of kimberlites. This study draws attention and can trigger further examinations into the nature of mantle metasomatism in lithospheric mantle using the composition of mantle minerals found in kimberlites specifically on the nature of fluid- vs. melt-driven metasomatism and its link to the origin of kimberlites. The practical application for exploration gives an additional bonus to this work. I think this study is of interest and significance for a large field of kimberlite and mantle studies.

The manuscript is very well written and presented. I only have few small comments/edits outlined below:

Line 185 – the best prove of significant diamond dissolution during residence in the mantle source is demonstrated by cathodoluminescence growth patterns. CL images of almost all diamonds show several episodes of significant resorption followed by re-growth. I believe the first study that draw attention to this was by Bulanova (1995). I suggest mentioning this evidence here.

Line 185 – it is better to say “occur in the mantle source” since most of kimberlite ascent is also happening in the mantle. This would avoid ambiguity in differentiating between the resorption by kimberlite vs. due to mantle metasomatism.

Line 186 – “is supported by...”

Line 188 – in addition to study (42) PhD thesis of Robinson (1979) also describes significant resorption of diamonds recovered from inside mantle xenoliths. Could be useful to add. While images in Robinson’s thesis are not clear, he gives detailed descriptions.

Line 234 – Leslie is a unique kimberlite in Lac de Gras, which is filled with magmatic kimberlite and interpreted to erupt less explosively (Nowicki et al 2008), probably because of the low content of volatiles. Diamonds from Leslie demonstrate resorption in volatile-undersaturated melt (Fedortchouk et al, 2010). The corrosive resorption style of diamonds and absence of fluid would suggest significant grade loss during kimberlite ascent. This is probably the most likely reason rather than sampling different lithosphere than the neighbouring kimberlites. Grizzly kimberlite probably shows the same.

Reviewer #2 (Remarks to the Author):

Review of Giuliani et al. submitted to Nature Communication 2023

The authors proposed that olivine geochemistry can be a new tool in diamond evaluation, namely predicting diamond grades, which is a complementary approach to the traditional evaluations based on the compositions of mantle-derived minerals such as garnet, chromite and clinopyroxene. Considering that olivine is the main constituent of fresh kimberlite rocks, it will have a powerful and widespread application in diamond exploration. I find that the study was carefully crafted and that the manuscript was well written. I therefore strongly recommend its publication in Nature Communication. Below I bring several points to the authors to consider.

1. The authors suggested that olivines with xenocrystic cores of high Mg# were thought to be minimally affected by kimberlite-related metasomatism and can be an indicator of good diamond preservation in the mantle lithosphere. This result is mainly based on the correlations between the composition of olivine and diamond grades in kimberlites worldwide. However, this result lacks further examination, such as by comparing the olivine compositions in kimberlites with that of olivine inclusions in diamond, for the latter undoubtedly avoids kimberlite-related metasomatism after diamond formation. Olivine inclusions in diamonds have $Mg\# \geq 90.1$ (Stachel and Harris, 2008 Ore Geology Reviews), which is consistent with the result in this study that $Mg\# \geq 90.3$ in cores of olivine from kimberlites with high diamond grades (≥ 50 cph). The authors should add the comparison content.

2. Olivine composition with low Mg# was used as a geochemical approach to show metasomatism, thus explaining the diamond grade in kimberlite. The variation of low-Mg# content in olivine should be mutually verified with the metasomatism shown by the mantle-derived indicator minerals recovered from the same kimberlite, such as whether the Mg# of olivine shows coordinated variation with that of Zr, Ti and other elements in garnet. The data of Zr and Ti content of garnet xenocrysts from different kimberlite should be collected and illustrated with Mg# of olivine suffered from metasomatism. It's not good enough for the simple description shown in lines 166-170.

3. In Lines 197-200, the authors suggested that kimberlites with high Mg# show higher proportions of octahedral diamond, and thus high diamond grades. Actually, the dissolution features of the diamond have a complex relationship with the diamond grade of kimberlite. Zhu et al. (2022, *Mineralium Deposita*) reported the distribution of major morphological forms of diamond in Wafangdian (previously call Fuxian) and Mengyin kimberlites, and found that kimberlites with high diamond grades such as have higher secondary dodecahedron form and less primary octahedron forms. So here the authors are advised to carefully check the related content.

REVISION NOTES

Reviewer #1 (Remarks to the Author): anonymous

This study reveals an interesting connection between the composition of olivine rims from kimberlites and diamond grade, connecting both to the extent of mantle metasomatism by proto-kimberlitic melts. The authors compiled a very large database to support their proposal and the conclusions are based on examination of this very large set of samples. Rigorous methods are used to eliminate or minimize biases in olivine analyses. The study introduces method of scoring of Mg-number of olivine cores and rims, and then combining them. Fig 4d is a very innovative way of correlating olivine composition with diamond grade. To the best of my knowledge, this is the first study that attempts to quantify the effect of partial dissolution of diamond due to mantle metasomatism on diamond grade of kimberlites. This study draws attention and can trigger further examinations into the nature of mantle metasomatism in lithospheric mantle using the composition of mantle minerals found in kimberlites specifically on the nature of fluid- vs. melt-driven metasomatism and its link to the origin of kimberlites. The practical application for exploration gives an additional bonus to this work. I think this study is of interest and significance for a large field of kimberlite and mantle studies.

We are pleased that this reviewer finds the conclusions of our study compelling and anticipates its broader application to understand mantle metasomatism and kimberlite genesis.

The manuscript is very well written and presented. I only have few small comments/edits outlined below:

Line 185 – the best prove of significant diamond dissolution during residence in the mantle source is demonstrated by cathodoluminescence growth patterns. CL images of almost all diamonds show several episodes of significant resorption followed by re-growth. I believe the first study that draw attention to this was by Bulanova (1995). I suggest mentioning this evidence here.

We prefer to not modify the manuscript here because cathodoluminescence imaging of diamonds provide evidence of multiple cycles of resorption and (over)growth during mantle residence and potentially in ancient times, whereas in this paragraph we address resorption by kimberlite-related melts or fluids followed by transport to the surface. We believe that, while the suggestion is valid, adding this information here might cause confusion to the reader.

Line 185 – it is better to say “occur in the mantle source” since most of kimberlite ascent is also happening in the mantle. This would avoid ambiguity in differentiating between the resorption by kimberlite vs. due to mantle metasomatism.

We have accepted this suggestion and amended the manuscript accordingly.

Line 186 – “is supported by...”

Amended.

Line 188 – in addition to study (42) PhD thesis of Robinson (1979) also describes significant resorption of diamonds recovered from inside mantle xenoliths. Could be useful to add.

While images in Robinson’s thesis are not clear, he gives detailed descriptions.

We have included a reference to Robinson (1979 PhD Thesis) at line 191.

Line 234 – Leslie is a unique kimberlite in Lac de Gras, which is filled with magmatic kimberlite and interpreted to erupt less explosively (Nowicki et al 2008), probably because of the low content of volatiles. Diamonds from Leslie demonstrate resorption in volatile-undersaturated melt (Fedortchouk et al, 2010). The corrosive resorption style of diamonds and absence of fluid would suggest significant grade loss during kimberlite ascent. This is probably the most likely reason rather than sampling different lithosphere than the neighbouring kimberlites. Grizzly kimberlite probably shows the same.

We have included this alternative hypothesis in the revised manuscript including a reference to Fedortchouk et al. (2010 EPSL) (lines 242-243).

Reviewer #2 (Remarks to the Author): Pei NI

Review of Giuliani et al. submitted to Nature Communication 2023

The authors proposed that olivine geochemistry can be a new tool in diamond evaluation, namely predicting diamond grades, which is a complementary approach to the traditional evaluations based on the compositions of mantle-derived minerals such as garnet, chromite and clinopyroxene. Considering that olivine is the main constituent of fresh kimberlite rocks, it will have a powerful and widespread application in diamond exploration. I find that the study was carefully crafted and that the manuscript was well written. I therefore strongly recommend its publication in Nature Communication. Below I bring several points to the authors to consider.

We are pleased that our manuscript was received positively by Dr Ni and he is supportive of publication.

1. The authors suggested that olivines with xenocrystic cores of high Mg# were thought to be minimally affected by kimberlite-related metasomatism and can be an indicator of good diamond preservation in the mantle lithosphere. This result is mainly based on the correlations between the composition of olivine and diamond grades in kimberlites worldwide. However, this result lacks further examination, such as by comparing the olivine compositions in kimberlites with that of olivine inclusions in diamond, for the latter undoubtedly avoids kimberlite-related metasomatism after diamond formation. Olivine inclusions in diamonds have $Mg\# \geq 90.1$ (Stachel and Harris, 2008 Ore Geology Reviews), which is consistent with the result in this study that $Mg\# \geq 90.3$ in cores of olivine from kimberlites with high diamond grades (≥ 50 cph). The authors should add the comparison content.

We have compared the average Mg# of olivine cores and rims from this study with that of olivine inclusions in diamonds using the compilation of Stachel et al. (2022 Rev Mineral Geochem). In updated Extended Data Table 1 we report average Mg# values of olivine inclusions in diamonds only for kimberlites for which at least 5 olivine compositions are documented. No correlation is observed (Revision Notes Figure 1). This is not unexpected because, as noted by Dr Ni, olivine included in diamonds are shielded from the effects of kimberlite-related metasomatism. In the revised manuscript we have included a statement that high Mg# is typical of olivine included in diamonds (lines 77-79).

Revision Notes Figure 1. Relationship between average Mg# of olivine inclusions in diamonds and average Mg# of olivine rims and cores in kimberlites.

2. Olivine composition with low Mg# was used as a geochemical approach to show metasomatism, thus explaining the diamond grade in kimberlite. The variation of low-Mg# content in olivine should be mutually verified with the metasomatism shown by the mantle-derived indicator minerals recovered from the same kimberlite, such as whether the Mg# of olivine shows coordinated variation with that of Zr, Ti and other elements in garnet. The data of Zr and Ti content of garnet xenocrysts from different kimberlite should be collected and illustrated with Mg# of olivine suffered from metasomatism. It's not good enough for the simple description shown in lines 166-170.

The suggestion of Dr Ni is valid. Indeed, in the early stages of this study we collated garnet major and trace element data for 26 kimberlites (data sources: Kobussen et al., 2009 Lithos; Griffin, W.L., unpublished data) for which olivine data are also available. No correlation is observed between average olivine core or rim Mg# and average garnet compositions, including Zr and Ti concentrations (Revision Notes Figure 2). A possible reason for this lack of correlation is that olivine derives from both garnet-bearing and garnet-free lithologies. While an interesting issue, we prefer exclude this discussion in the revised manuscript, because part of the garnet data we have employed are not published and, in addition, this (unsuccessful) comparison does not add any value to the manuscript contents. We agree that it will be an interesting avenue for future research and hope that our paper will stimulate this line of research.

Revision Notes Figure 2. Relationship between average Mg# of olivine cores and average compositions of garnet xenocrysts in kimberlites.

3. In Lines 197-200, the authors suggested that kimberlites with high Mg# show higher proportions of octahedral diamond, and thus high diamond grades. Actually, the dissolution features of the diamond have a complex relationship with the diamond grade of kimberlite. Zhu et al. (2022, Mineralium Deposita) reported the distribution of major morphological forms of diamond in Wafangdian (previously call Fuxian) and Mengyin kimberlites, and found that kimberlites with high diamond grades have higher secondary dodecahedron form and less primary octahedron forms. So here the authors are advised to carefully check the related content.

We thank the reviewer for drawing out attention to this issue. We have revised the manuscript to include the observations of Zhu and co-workers which further confirm that additional work will be required to assess the potential relationship between diamond rounding, diamond resorption features, and olivine composition (lines 203-206).

REVIEWERS' COMMENTS

Reviewer #2 (Remarks to the Author):

After review of the revised manuscript and the author's responses to the previous review comments, I think the author responded well to the questions raised and made changes accordingly. The quality of the present revised manuscript has met the requirements for publication, and it is recommended to accept the manuscript for publication.